# Training Agent for First-Person Shooter Game with Actor-Critic Curriculum Learning

**Yuxin Wu**
Carnegie Mellon University
ppwwyyxx@gmail.com

**Yuandong Tian**
Facebook AI Research
yuandong@fb.com

## Abstract

In this paper, we propose a new framework for training vision-based agent for First-Person Shooter (FPS) Game, in particular Doom. Our framework combines the state-of-the-art reinforcement learning approach (Asynchronous Advantage Actor-Critic (A3C) model [Mnih et al. (2016)]) with curriculum learning. Our model is simple in design and only uses game states from the AI side, rather than using opponents' information [Lample & Chaplot (2016)]. On a known map, our agent won 10 out of the 11 attended games and the champion of Track1 in ViZ-Doom AI Competition 2016 by a large margin, 35% higher score than the second place.

## 1 Introduction

Deep Reinforcement Learning has achieved super-human performance in fully observable environments, e.g., in Atari Games [Mnih et al. (2015)] and Computer Go [Silver et al. (2016)]. Recently, Asynchronous Advantage Actor-Critic (A3C) [Mnih et al. (2016)] model shows good performance for 3D environment exploration, e.g. labyrinth exploration. However, in general, to train an agent in a partially observable 3D environment from raw frames remains an open challenge. Direct application of A3C to competitive 3D scenarios, e.g. 3D games, is nontrivial, partly due to sparse and long-term rewards in such scenarios.

Doom is a 1993 First-Person Shooter (FPS) game in which a player fights against other computer-controlled agents or human players in an adversarial 3D environment. Previous works on FPS AI [van Waveren (2001)] focused on using hand-tuned state machines and privileged information, e.g., the geometry of the map, the precise location of all players, to design playable agents. Although state-machine is conceptually simple and computationally efficient, it does not operate like human players, who only rely on visual (and possibly audio) inputs. Also, many complicated situations require manually-designed rules which could be time-consuming to tune.

In this paper, we train an AI agent in Doom with a framework that based on A3C with convolutional neural networks (CNN). This model uses only the recent 4 frames and game variables from the AI side, to predict the next action of the agent and the value of the current situation. We follow the curriculum learning paradigm [Bengio et al. (2009); Jiang et al. (2015)]: start from simple tasks and then gradually try harder ones. The difficulty of the task is controlled by a variety of parameters in Doom environment, including different types of maps, strength of the opponents and the design of the reward function. We also develop adaptive curriculum training that samples from a varying distribution of tasks to train the model, which is more stable and achieves higher score than A3C with the same number of epoch. As a result, our trained agent, named *F1*, won the champion in Track 1 of ViZDoom Competition [1] by a large margin.

There are many contemporary efforts on training a Doom AI based on the VizDoom platform [Kempka et al. (2016)] since its release. Arnold [Lample & Chaplot (2016)] also uses game frames and trains an action network using Deep Recurrent Q-learning [Hausknecht & Stone (2015)], and a navigation network with DQN [Mnih et al. (2015)]. However, there are several important differences. To predict the next action, they use a hybrid architecture (CNN+LSTM) that involves more complicated training procedure. Second, in addition to game frames, they require internal

---

[1] http://vizdoom.cs.put.edu.pl/competition-cig-2016/results

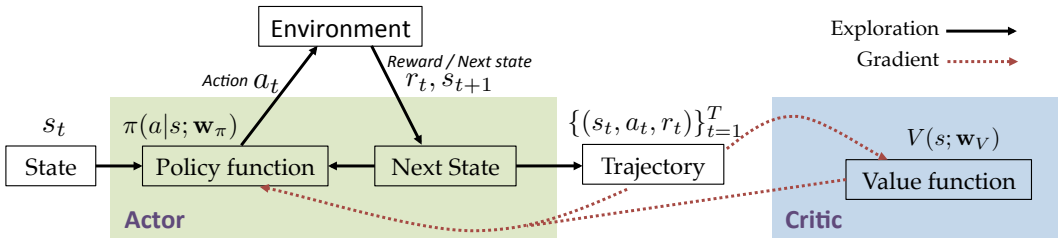

Figure 1: The basic framework of actor-critic model.

game status about the opponents as extra supervision during training, e.g., whether enemy is present in the current frame. IntelAct [Dosovitskiy & Koltun (2017)] models the Doom AI bot training in a supervised manner by predicting the future values of game variables (e.g., health, amount of ammo, etc) and acting accordingly. In comparison, we use curriculum learning with asynchronized actor-critic models and use stacked frames (4 most recent frames) and resized frames to mimic short-term memory and attention. Our approach requires no opponent's information, and is thus suitable as a general framework to train agents for close-source games.

In VizDoom AI Competition 2016 at IEEE Computational Intelligence And Games (CIG) Conference[2], our AI won the champion of Track1 (limited deathmatch with known map), and IntelAct won the champion of Track2 (full deathmatch with unknown maps). Neither of the two teams attends the other track. Arnold won the second places of both tracks and CLYDE [Ratcliffe et al. (2017)] won the third place of Track1.

## 2 THE ACTOR-CRITIC MODEL

The goal of Reinforcement Learning (RL) is to train an agent so that its behavior maximizes/minimizes expected future rewards/penalties it receives from a given environment [Sutton & Barto (1998)]. Two functions play important roles: a value function $V(s)$ that gives the expected reward of the current state $s$, and a policy function $\pi(a|s)$ that gives a probability distribution on the candidate actions $a$ for the current state $s$. Getting the groundtruth value of either function would largely solve RL: the agent just follows $\pi(a|s)$ to act, or jumps in the best state provided by $V(s)$ when the number of candidate next states is finite and practically enumerable. However, neither is trivial.

Actor-critic models [Barto et al. (1983); Sutton (1984); Konda & Tsitsiklis (1999); Grondman et al. (2012)] aim to jointly estimate $V(s)$ and $\pi(a|s)$: from the current state $s_t$, the agent explores the environment by iteratively sampling the policy function $\pi(a_t|s_t; \mathbf{w}_\pi)$ and receives positive/negative reward, until the terminal state or a maximum number of iterations are reached. The exploration gives a trajectory $\{(s_t, a_t, r_t), (s_{t+1}, a_{t+1}, r_{t+1}), \cdots\}$, from which the policy function and value function are updated. Specifically, to update the value function, we use the expected reward $R_t$ along the trajectory as the ground truth; to update the policy function, we encourage actions that lead to high rewards, and penalize actions that lead to low rewards. To determine whether an action leads to high- or low-rewarding state, a reference point, called *baseline* [Williams (1992)], is usually needed. Using zero baseline might increase the estimation variance. [Peters & Schaal (2008)] gives a way to estimate the best baseline (a weighted sum of cumulative rewards) that minimizes the variance of the gradient estimation, in the scenario of episodic REINFORCE [Williams (1992)].

In actor-critic frameworks, we pick the baseline as the expected cumulative reward $V(s)$ of the current state, which couples the two functions $V(s)$ and $\pi(a|s)$ together in the training, as shown in Fig. 1. Here the two functions reinforce each other: a correct $\pi(a|s)$ gives high-rewarding trajectories which update $V(s)$ towards the right direction; a correct $V(s)$ picks out the correct actions for $\pi(a|s)$ to reinforce. This mutual reinforcement behavior makes actor-critic model converge faster, but is also prone to converge to bad local minima, in particular for on-policy models that follow the very recent policy to sample trajectory during training. If the experience received by the agent in consecutive batches is highly correlated and biased towards a particular subset of the environment, then both $\pi(a|s)$ and $V(s)$ will be updated towards a biased direction and the agent may never see

---

[2]http://vizdoom.cs.put.edu.pl/competition-cig-2016

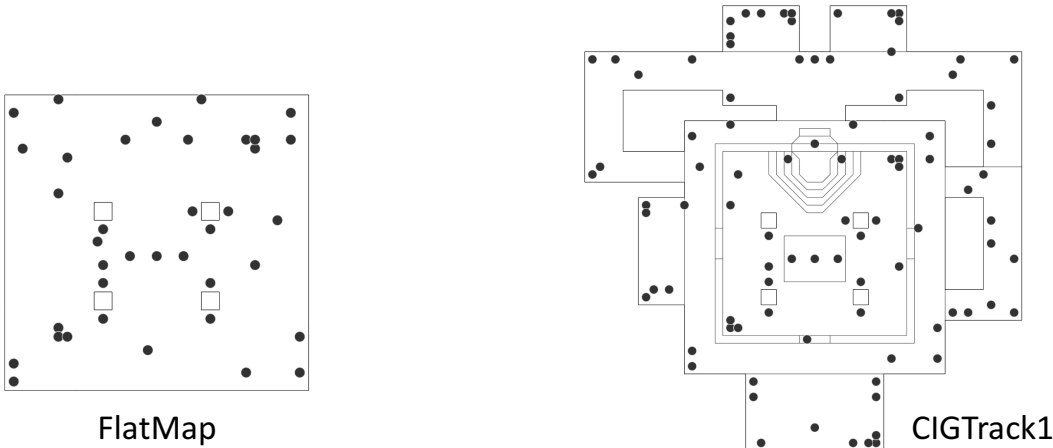

Figure 2: Two maps we used in the paper. `FlatMap` is a simple square containing four pillars . `CIGTrack1` is the map used in Track1 in ViZDoom AI Competition (We did not attend Track2). Black dots are items (weapons, ammo, medkits, armors, etc).

the whole picture. To reduce the correlation of game experience, Asynchronous Advantage Actor-Critic Model [Mnih et al. (2016)] runs independent multiple threads of the game environment in parallel. These game instances are likely uncorrelated, therefore their experience in combination would be less biased.

For on-policy models, the same mutual reinforcement behavior will also lead to highly-peaked $\pi(a|s)$ towards a few actions (or a few fixed action sequences), since it is always easy for both actor and critic to over-optimize on a small portion of the environment, and end up "living in their own realities". To reduce the problem, [Mnih et al. (2016)] added an entropy term to the loss to encourage diversity, which we find to be critical. The final gradient update rules are listed as follows:

$$\mathbf{w}_\pi \leftarrow \mathbf{w}_\pi + \alpha(R_t - V(s_t))\nabla_{\mathbf{w}_\pi} \log \pi(a_t|s_t) + \beta \nabla_{\mathbf{w}_\pi} H(\pi(\cdot|s_t)) \tag{1}$$

$$\mathbf{w}_V \leftarrow \mathbf{w}_V - \alpha \nabla_{\mathbf{w}_V} (R_t - V(s_t))^2 \tag{2}$$

where $R_t = \sum_{t'=t}^{T} \gamma^{t'-t} r_{t'}$ is the expected discounted reward at time $t$ and $\alpha$, $\beta$ are the learning rate. In this work, we use Huber loss instead of the L2 loss in Eqn. 2.

**Architecture.** While [Mnih et al. (2016)] keeps a separate model for each asynchronous agent and perform model synchronization once in a while, we use an alternative approach called Batch-A3C, in which all agents act on the same model and send batches to the main process for gradient descent optimization. The agents' models are updated after each gradient update. Note that the contemporary work GA3C [Babaeizadeh et al. (2017)] also proposes a similar architecture. In their architecture, there is a prediction queue that collects agents' experience and sends them to multiple predictors, and a training queue that collects experience to feed the optimization.

## 3 DOOM AS A REINFORCEMENT LEARNING PLATFORM

In Doom, the player controls the agent to fight against enemies in a 3D environment (e.g., in a maze). The agent can only see the environment from his viewpoint and thus receives partial information upon which it makes decisions. On modern computers, the original Doom runs in thousands of frames per second, making it suitable as a platform for training AI agent. ViZDoom [Kempka et al. (2016)] is an open-source platform that offers programming interface to communicate with Doom engine, ZDoom[3]. From the interface, users can obtain current frames of the game, and control the agent's action. ViZDoom offers much flexibility, including:

*Rich Scenarios.* Many customized scenarios are made due to the popularity of the game, offering a variety of environments to train from. A scenario consists of many components, including 2D maps for the environment, scripts to control characters and events. Open-source tools, such as

---

[3]`https://zdoom.org/`

SLADE[4], are also widely available to build new scenarios. We built our customized map (Fig. 2(b)) for training.

*Game variables.* In addition to image frames, ViZDoom environment also offers many games variables revealing the internal state of the game. This includes `HEALTH`, `AMMO_?` (agent's health and ammunition), `FRAG_COUNT` (current score) and so on. ViZDoom also offers `USER?` variables that are computed on the fly via scenario scripts. These `USER?` variables can provide more information of the agent, e.g., their spatial locations. Enemy information could also be obtained by modifying ViZDoom [Lample & Chaplot (2016)]. Such information is used to construct a reward function, or as a direct supervision to accelerate training [Lample & Chaplot (2016)].

*Built-in bots.* Built-in bots can be inserted in the battle. They are state machines with privileged information over the map and the player, which results in apparently decent intelligence with minimal computational cost. By competing against built-in bots, the agent learns to improve.

*Evaluation Criterion.* In FPS games, to evaluate their strength, multiple AIs are placed to a scenario for a *deathmatch*, in which every AI plays for itself against the remaining AIs. *Frags* per episode, the number of kills minus the number of suicides for the agent in one round of game, is often used as a metric. An AI is stronger if its frags is ranked higher against others. In this work, we use an episode of 2-minute game time (4200 frames in total) for all our evaluations unless noted otherwise.

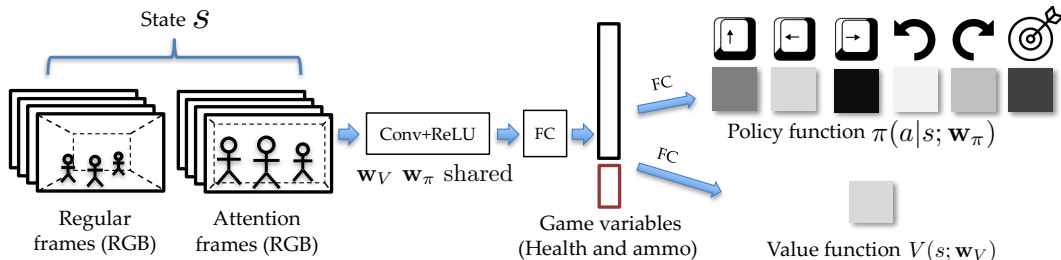

Figure 3: The network structure of the proposed model. It takes 4 recent game frames plus 4 recent attention frames as the input state $s$, and outputs a probability distribution $\pi(a|s)$ of the 6 discrete actions. The policy and value network share parameters.

## 4 METHOD

### 4.1 NETWORK ARCHITECTURE

We use convolutional neural networks to extract features from the game frames and then combine its output representation with game variables. Fig. 3 shows the network architecture and Tbl. 1 gives the parameters. It takes the frames as the input (i.e., the state $s$) and outputs two branches, one that outputs the value function $V(s)$ by regression, while the other outputs the policy function $\pi(s|a)$ by a regular softmax. The parameters of the two functions are shared before the branch.

For input, we use the most recent 4 frames plus the center part of them, scaled to the same size ($120 \times 120$). Therefore, these centered "attention frames" have higher resolution than regular game frames, and greatly increase the aiming accuracy. The policy network will give 6 actions, namely `MOVE_FORWARD`, `MOVE_LEFT`, `MOVE_RIGHT`, `TURN_LEFT`, `TURN_RIGHT`, and `ATTACK`. We found other on-off actions (e.g., `MOVE_BACKWARD`) offered by ViZDoom less important. After feature extraction by convolutional network, game variables are incorporated. This includes the agent's Health (0-100) and Ammo (how many bullets left). They are related to AI itself and thus legal in the game environment for training, testing and ViZDoom AI competition.

### 4.2 TRAINING PIPELINE

Our training procedure is implemented with TensorFlow [Abadi et al. (2016)] and tensorpack[5]. We open 255 processes, each running one Doom instance, and sending experience $(s_t, a_t, r_t)$ to the

---

[4]`http://slade.mancubus.net/`
[5]`https://github.com/ppwwyyxx/tensorpack`

| Layer # | 1 | 2 | 3 | 4 | 5 | 6 | 7 |
|---------|---|---|---|---|---|---|---|
| | C7x7x32s2 | C7x7x64s2 | MP3x3s2 | C3x3x128 | MP3x3s2 | C3x3x192 | FC1024 |

Table 1: Network parameters. *C7x7x32s2* = convolutional layer with 7x7 kernel, stride 2 and number of output planes 32. *MP* = MaxPooling. Each convolutional and fully connected layer is followed by a ReLU, except for the last output layer.

| Parameters | Description | FlatMap | CIGTrack1 |
|------------|-------------|---------|-----------|
| living | Penalize agent who just lives | -0.008 / action | |
| health_loss | Penalize health decrement | -0.05 / unit | |
| ammo_loss | Penalize ammunition decrement | -0.04 / unit | |
| health_pickup | Reward for medkit pickup | 0.04 / unit | |
| ammo_pickup | Reward for ammunition pickup | 0.15 / unit | |
| dist_penalty | Penalize the agent when it stays | -0.03 / action | |
| dist_reward | Reward the agent when it moves | 9e-5 / unit distance | |
| dist_penalty_thres | Threshold of displacement | 8 | 15 |
| num_bots | Number of built-in bots | 8 | 16 |

Table 2: Parameters for different maps.

main process which runs the training procedure. The main process collects frames from different game instances to create batches, and optimizes on these batches asynchronously on one or more GPUs using Eqn. 1 and Eqn. 2. The frames from different processes running independent game instances, are likely to be uncorrelated, which stabilizes the training. This procedure is slightly different from the original A3C, where each game instance collects their own experience and updates the parameters asynchronously.

Despite the use of entropy term, we still find that $\pi(\cdot|s)$ is highly peaked. Therefore, during trajectory exploration, we encourage exploration by the following changes: a) multiply the policy output of the network by an exploration factor (0.2) before softmax b) uniformly randomize the action for 10% random frames.

As mentioned in [Kempka et al. (2016)], care should be taken for frame skips. Small frame skip introduces strong correlation in the training set, while big frame skip reduces effective training samples. We set frame skip to be 3. We choose 640x480 as the input frame resolution and do not use high aspect ratio resolution [Lample & Chaplot (2016)] to increase the field of view.

We use Adam [Kingma & Ba (2014)] with $\epsilon = 10^{-3}$ for training. Batch size is 128, discount factor $\gamma = 0.99$, learning rate $\alpha = 10^{-4}$ and the policy learning rate $\beta = 0.08\alpha$. The model is trained from scratch. The training procedure runs on Intel Xeon CPU E5-2680v2 at 2. 80GHz, and 2 TitanX GPUs. It takes several days to obtain a decent result. Our final model, namely the *F1* bot, is trained for around 3 million mini-batches on multiple different scenarios.

## 4.3 CURRICULUM LEARNING

When the environment only gives very sparse rewards, or adversarial, A3C takes a long time to converge to a satisfying solution. A direct training with A3C on the map CIGTrack1 with 8 built-in bots does not yield sensible performance. To address this, we use *curriculum learning* [Bengio et al. (2009)] that trains an agent with a sequence of progressively more difficult environments. By varying parameters in Doom (Sec. 3), we could control its difficulty level.

| | Class 0 | Class 1 | Class 2 | Class 3 | Class 4 | Class 5 | Class 6 | Class 7 |
|---|---------|---------|---------|---------|---------|---------|---------|---------|
| Speed | 0.2 | 0.2 | 0.4 | 0.4 | 0.6 | 0.8 | 0.8 | 1.0 |
| Health | 40 | 40 | 40 | 60 | 60 | 60 | 80 | 100 |

Table 3: Curriculum design for FlatMap. Note that enemy uses RocketLauncher except for Class 0 (Pistol).

**Reward Shaping.** Reward shaping has been shown to be an effective technique to apply reinforcement learning in a complicated environment with delayed reward [Ng et al. (1999); Devlin et al. (2011)]. In our case, besides the basic reward for kills (+1) and death (-1), intermediate rewards are used as shown in Tbl. 2. We penalize agent with a living state, encouraging it to explore and encounter more enemies. `health_loss` and `ammo_loss` place linear reward for a decrement of health and ammunition. `ammo_pickup` and `health_pickup` place reward for picking up these two items. In addition, there is extra reward for picking up ammunition when in need (e.g. almost out of ammo). `dist_penalty` and `dist_reward` push the agent away from the previous locations, encouraging it to explore. The penalty is applied every action, when the displacement of the bot relative to the last state is less than a threshold `dist_penalty_thres`. And `dist_reward` is applied for every unit displacement the agent makes. Similar to [Lample & Chaplot (2016)], the displacement information is computed from the ground truth location variables provided by Doom engine, and will not be used in the competition. However, unlike [Lample & Chaplot (2016)] that uses enemy-in-sight signal for training, locations can be extracted directly from `USER?` variables, or can easily be computed roughly with action history.

**Curriculum Design.** We train the bot on `FlatMap` that contains a simple square with a few pillars (Fig. 2(a)) with several curricula (Tbl. 3), and then proceed to `CIGTrack1`. For each map, we design curricula by varying the strength of built-in bots, i.e., their moving speed, initial health and initial weapon. Our agent always uses `RocketLauncher` as its only weapon. Training on `FlatMap` leads to a capable initial model which is quickly adapted to more complicated maps. As shown in Tbl. 2, for `CIGTrack1` we increase `dist_penalty_thres` to keep the agent moving, and increase `num_bots` so that the agent encounters more enemies per episode.

**Adaptive Curriculum.** In addition to staged curriculum learning, we also design adaptive curriculum learning by assigning a probability distribution on different levels for each thread that runs a Doom instance. The probability distribution shifts towards more difficult curriculum when the agent performs well on the current distribution, and shifts towards easier level otherwise. We consider the agent to perform well if its frag count is greater than 10 points.

## 4.4 Post-training Rules

For a better performance in the competition, we also put several rules to process the action given by the trained policy network, called post-training (PT) rules. There are two sets of buttons in ViZDoom: on-off buttons and delta buttons. While on-off button maps to the binary states of a keystroke (e.g., pressing the up arrow key will move the agent forward), delta buttons mimic the mouse behavior and could act faster in certain situations. Therefore, we setup rules that detect the intention of the agent and accelerate with delta button. For example, when the agent turns by invoking `TURN_LEFT` repeatedly, we convert its action to `TURN_LEFT_RIGHT_DELTA` for acceleration. Besides, the trained model might get stuck in rare situations, e.g., keep moving forward but blocked by an explosive bucket. We also designed rules to detect and fix them.

## 5 Experiment

In this section, we show the training procedure (Sec. 5.1), evaluate our AIs with ablation analysis (Sec. 5.2) and ViZDoom AI Competition (Sec. 5.3). We mainly compare among three AIs: (1) *F1Pre*, the bot trained with `FlatMap` only, (2) *F1Plain*, the bot trained on both `FlatMap` and `CIGTrack1`, but without post-training rules, and (3) the final *F1* bot that attends competition.

## 5.1 Curriculum Learning on FlatMap

Fig. 4 shows that the curriculum learning increases the performance of the agents over all levels. When an agent becomes stronger in the higher level of class, it is also stronger in the lower level of class without overfitting. Fig. 5 shows comparison between adaptive curriculum learning with pure A3C. We can see that pure A3C can learn on `FlatMap` but is slower. Moreover, in `CIGTrack1`, a direct application of A3C does not yield sensible performance.

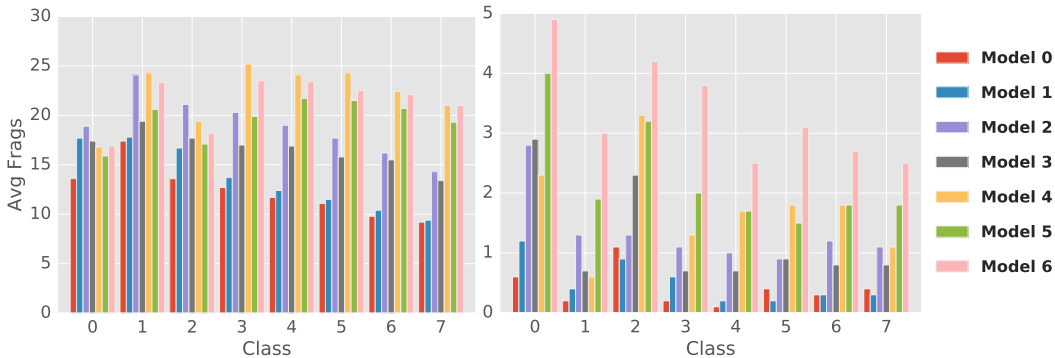

Figure 4: Average Frags over 300 episodes evaluation, on `FlatMap`(left) and `CIGTrack1`(right) with different levels of enemies (See Tbl. 3 for curriculum design). Models from later stages performs better especially on the difficult map, yet still keeps a good performance on the easier map.

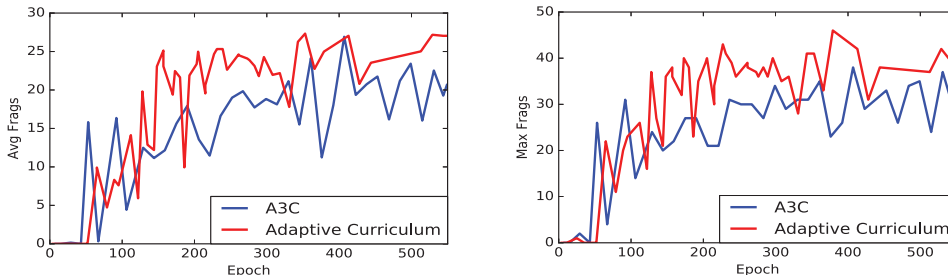

Figure 5: Performance comparison on Class 7 (hardest) of `FlatMap` between A3C [Mnih et al. (2016)] and adaptive curriculum learning, at different stage of training. Average frags and max frags are computed from 100 episodes. Adaptive curriculum shows higher performance and is relatively more stable.

## 5.2 ABLATION ANALYSIS

**Visualization.** Fig. 6 shows the visualization of the first convolutional layer of the trained AI agent. We could see that the convolutional kernels of the current frame is less noisy than the kernels of previous frames. This means that the agent makes the most use of the current frames.

**Effect of History Frames.** Interestingly, while the agent focuses on the current frame, it also uses motion information. For this, we use (1) 4 duplicated current frames (2) 4 recent frames in reverse order, as the input. This gives 8.50 and 2.39 mean frags, compared to 10.34 in the normal case, showing that the agent heavily uses the motion information for better decision. In particular, the bot is totally confused with the reversed motion feature. Detailed results are shown in Tbl. 5.

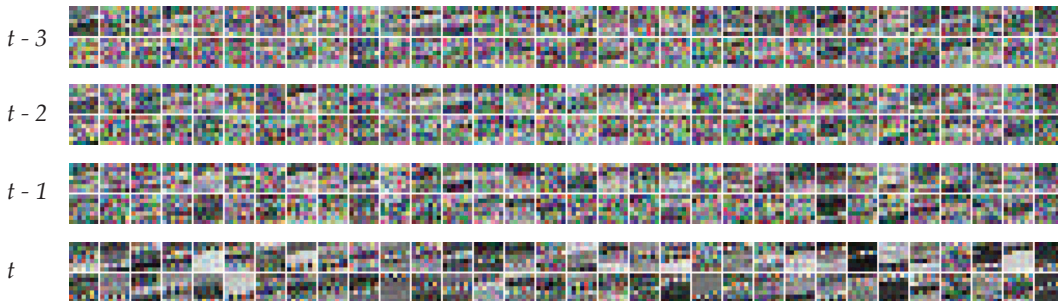

Figure 6: Visualization of the convolutional filters in the first layer of our network. The filters are grouped by the frame index they corresponds to. Each group consists of two rows of 32 RGB filters for the regular and attention frames, respectively. The filters corresponding to the current frame (last row) is less noisy than those of others, showing that the bot is more focused on the current frame.

| | Built-In AI | F1Pre | F1Plain | F1 |
|---|---|---|---|---|
| FlatMap | 8.07/20 | 14.47/24 | 17.26/29 | 22.45/37 |
| CIGTrack1 | 0.48/7 | 3.56/15 | 8.58/16 | 10.65/18 |

Table 4: Avg/Max frags of each AIs in the internal tournament (150 episodes of 10 minutes each).

| | FlatMap | | | CIGTrack1 | | |
|---|---|---|---|---|---|---|
| | Min | Mean | Max | Min | Mean | Max |
| F1 bot (reverse history) | 1 | 9.89 | 19 | -2 | 2.39 | 9 |
| F1 bot (duplicated history) | 10 | 24.62 | 37 | 2 | 8.50 | 17 |
| F1 bot (w/o PT rules) | 14 | 22.80 | 36 | 1 | 8.66 | 18 |
| F1 bot | 16 | 25.17 | 37 | 5 | 10.34 | 17 |

Table 5: Performance evaluation (in terms of frags) on two standard scenarios `FlatMap` and `CIGTrack1` over 300 episodes. Our bot performs better with post-training rules.

**Post-training Rules.** Tbl. 5 shows that the post-training rules improve the performance. As a future work, an end-to-end training involving delta buttons could make the bot better.

**Internal Tournament.** We also evaluate our AIs with internal tournaments (Tbl. 4). All our bots beat the performance of built-in bots by a large margin, even though they use privileged information. *F1Pre*, trained with only `FlatMap`, shows decent performance, but is not as good as the models trained with both `FlatMap` and `CIGTrack1`. The final bot *F1* performs the best.

**Behaviors.** Visually, the three bots behave differently. *F1Pre* is a bit overtrained in `FlatMap` and does not move too often, but when it sees enemies, even faraway, it will start to shoot. Occasionally it will move to the corner and pick medkits. In `CIGTrack1`, *F1Pre* stays in one place and ambushes opponents who pass by. On the other hand, *F1Plain* and *F1* always move forwards and turn at the corner. As expected, *F1* moves and turns faster.

**Tactics** All bots develop interesting local tactics when exchanging fire with enemy: they slide around when shooting the enemy. This is quite effective for dodging others' attack. Also when they shoot the enemy, they usually take advantage of the splashing effect of rocket to cause additional damage for enemy, e.g., shooting the wall when the enemy is moving. They do not pick ammunition too often, even if they can no longer shoot. However, such disadvantage is mitigated by the nature of deathmatch: when a player dies, it will respawn with ammunition. We also check states with highest/lowest estimated future value $V(s)$ over a 10-episode evaluation of *F1* bot, from which we can speculate its tactics. The highest value is $V = 0.97$ when the agent fired, and about to hit the enemy. One low value is $V = -0.44, ammo = 0$, when the agent encountered an enemy at the corner but is out of ammunition. Both cases are reasonable.

## 5.3 COMPETITION

We attended the ViZDoom AI Competition hosted by IEEE CIG. There are 2 tracks in the competition. Track 1 (Limited Deathmatch) uses a known map and fixed weapons, while Track 2 (Full Deathmatch) uses 3 unknown maps and a variety of weapons. Each bot fights against all others for 12 rounds of 10 minutes each. Due to server capacity, each bot skips one match in the first 9 rounds. All bots are supposed to run in real-time (>35fps) on a GTX960 GPU.

| Round | 1 | 2 | 3 | 4 | 5 | 6 | 7 | 8 | 9 | 10 | 11 | 12 | Total |
|---|---|---|---|---|---|---|---|---|---|---|---|---|---|
| Our bot | **56** | **62** | n/a | **54** | **47** | 43 | **47** | **55** | **50** | **48** | **50** | **47** | **559** |
| Arnold | 36 | 34 | **42** | 36 | 36 | **45** | 36 | 39 | n/a | 33 | 36 | 40 | 413 |
| CLYDE | 37 | n/a | 38 | 32 | 37 | 30 | 46 | 42 | 33 | 24 | 44 | 30 | 393 |

Table 6: Top 3 teams in ViZDoom AI Competition, Track 1. Our bot attended 11 out of 12 games, won 10 of them and won the champion by a large margin. For design details, see Arnold [Lample & Chaplot (2016)] and CLYDE [Ratcliffe et al. (2017)].

Our *F1* bot won 10 out of 11 attended games and won the champion for Track 1 by a large margin. We have achieved 559 frags, 35.4% higher than 413 frags achieved by Arnold [Lample & Chaplot (2016)], that uses extra game state for model training. On the other hand, IntelAct [Dosovitskiy & Koltun (2017)] won Track 2. The full videos for the two tracks have been released[6][7], as well as an additional game between Human and AIs[8]. Our bot behaves reasonable and very human-like in Track 1. In the match between Human and AIs, our bot was even ahead of the human player for a short period (6:30 to 7:00).

## 6 CONCLUSION

Teaching agents to act properly in complicated and adversarial 3D environment is a very challenging task. In this paper, we propose a new framework to train a strong AI agent in a First-Person Shooter (FPS) game, Doom, using a combination of state-of-the-art Deep Reinforcement Learning and Curriculum Training. Via playing against built-in bots in a progressive manner, our bot wins the champion of Track1 (known map) in ViZDoom AI Competition. Furthermore, it learns to use motion features and build its own tactics during the game, which is never taught explicitly.

Currently, our bot is still an reactive agent that only remembers the last 4 frames to act. Ideally, a bot should be able to build a map from an unknown environment and localize itself, is able to have a global plan to act, and visualize its reasoning process. We leave them to future works.

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
