# Peer review of "Training Agent for First-Person Shooter Game with Actor-Critic Curriculum Learning"

_ICLR 2017 — accepted_

[Official Review · AnonReviewer3 · rating 4 · confidence 5 · 15 Dec 2016]
**No Title**

This paper basically applies A3C to 3D spatial navigation tasks. 

- This is not the first time A3C has been applied to 3D navigation. In fact the original paper reported these experiments. Although the experimental results are great, I am not sure if this paper has any additional insights to warrant itself as a conference paper. It might make more sense as a workshop paper

-  Are the graphs in Fig 5 constructed using a single hyper-parameter sweep? I think the authors should report results with many random initializations to make the comparisons more robust

- Overall the two main ideas in this paper -- A3C and curriculums -- are not really novel but the authors do make use of them in a real system.

[Official Review · AnonReviewer2 · rating 7 · confidence 4 · 15 Dec 2016]
**No Title**

This is a solid paper that applies A3C to Doom, enhancing it with a collection of tricks so as to win one of the VizDoom competitions. I think it is fair to expect the competition aspect to overshadow the more scientific approach of justifying every design decision in isolation, but in fact the authors do a decent job at the latter.

Two of my concerns have remained unanswered (see AnonReviewer2, below). 

In addition, the citation list is rather thin, for example reward shaping has a rich literature, as do incrementally more difficult task setups, dating back at least to Mark Ring’s work in the 1990s. There has also been a lot of complementary work on other FPS games. I’m not asking that the authors do any direct comparisons, but to give the reader a sense of context in which to place this.

[Official Review · AnonReviewer1 · rating 6 · confidence 4 · 19 Dec 2016 (modified: 23 Jan 2017)]
**Final Review: Practical techniques for learning in 3D shooters; a lot of domain knowledge, and a few iffy claims**

The paper describes approaches taken to train learning agents for the 3D game Doom. The authors propose a number of performance enhancements (curriculum learning, attention (zoomed-in centered) frames, reward shaping, game variables, post-training rules) inspired by domain knowledge.

The enhancements together lead to a clear win as demonstrated by the competition results. From Fig 4, the curriculum learning clearly helps with learning over increasingly difficult settings. A nice result is that there is no overfitting to the harder classes once they have learned (probably because the curriculum is health and speed). The authors conclude from Fig 5 that the adaptive curriculum is better and more stable that pure A3C; however, this is a bit of a stretch given that graph. They go on to say that Pure A3C doesn't learn at all in the harder map but then show no result/graph to back this claim. Tbl 5 shows a clear benefit of the post-training rules.

If the goal is to solve problems like these (3D shooters), then this paper makes a significant contribution in that it shows which techniques are practical for solving the problem and ultimately improving performance in these kinds of tasks. Still, I am just not excited about this paper, mainly because it relies so heavily of many sources of domain knowledge, it is quite far from the pure reinforcement learning problem. The results are relatively unsurprising. Maybe they are novel for this problem, though.

I'm not sure we can realistically draw any conclusions about Figure 6 in the paper's current form. I recommend the authors increase the resolution or run some actual metrics to determine the fuzziness/clarity of each row/image: something more concrete than an arrow of already low-resolution images.

--- Added after rebuttal:

I still do not see any high-res images for Figure 6 or any link to them, but I trust that the authors will add them if accepted.

[Author Response · Yuandong Tian · 14 Jan 2017]
**Rebuttal**

We thank the reviewers for their insightful comments!

All reviewers agree that this paper makes a solid contribution with good experimental results. It is not uncommon to see application-oriented papers using a combination of multiple techniques to achieve strong performance. This category covers many seminar works, e.g., deep reinforcement learning for Atari games (applying deep models to traditional Q-learning), or even AlphaGo (supervised learning, policy gradient, value function, Monte-Carlo Tree Search, self-play). It may be a bit shortsighted to judge such strong performing papers with a single criterion.

Confusion about the domain:
Reviewer3 mentions that the paper "basically applies A3C to 3D spatial navigation tasks.", which is not true. In the deathmatch game of Doom, multiple players explore the maze and fight against each other to get a higher score, which is defined as #kills - #suicide. In this task, part of the goal is to learn anti-enemy tactics (e.g., dodging the rocket shot from the enemy, e.g., video:

[Final Decision · Program Chairs · 06 Feb 2017]
**ICLR committee final decision**

This paper provides a number of performance enhancements inspired by domain knowledge. Taken together, these produce a compelling system that has shown itself to be the best-in-class as per the related competition.
 Experts agree that the authors do a good job at justifying the majority of the design decisions.
 
 pros:
 - insights into the SOTA Doom player
 
 cons:
 - lack of pure technical novelty: the various elements have existed previously
 
 This paper comes down to a matter of taste in terms of appreciation of SOTA systems or technical novelty.
 With the code being released, I believe that this work will have impact as a benchmark, and as a guidebook
 as to how features can be combined for SOTA performance on FPS-style scenarios.